# An Orthogonal Experimental Study on the Preparation of Cr Coatings on Long-Size Zr Alloy Tubes by Arc Ion Plating

**DOI:** 10.3390/ma15207177

**Published:** 2022-10-14

**Authors:** Huan Chen, Zhaodandan Ma, Yu Wang, Tianguo Wei, Hongyan Yang, Peinan Du, Xiaomin Wang, Ruiqian Zhang

**Affiliations:** 1Key Laboratory of Reactor Fuel and Materials, Chengdu 610213, China; 2Nuclear Power Institute of China, Chengdu 610213, China

**Keywords:** ATF, Zr alloy, Cr coating, arc ion plating, orthogonal analysis

## Abstract

Cr-coated Zr alloys are widely considered the most promising accident-tolerant fuel (ATF) cladding materials for engineering applications in the near term. In this work, Cr coatings were prepared on the surfaces of 1400 mm long N36 cladding tubes using an industrial multiple arc source system. Orthogonal analyses were conducted to demonstrate the significance level of various process parameters influencing the characteristics of coatings (surface roughness, defects, crystal orientation, grain structure, etc.). The results show that the arc current mainly affects the coating deposition rate and the droplet particles on the surface or inside the coatings; however, the crystal preferred orientation and grain structure are more significantly influenced by the gas pressure and negative bias voltage, respectively. Then, the underlying mechanisms are carefully discussed. At last, a set of systemic methods to control the quality and microstructures of Cr coatings are summarized.

## 1. Introduction

The research and development of Accident Tolerant Fuel (ATF) cladding materials were raised by the Fukushima nuclear accident in Japan due to nuclear safety concerns [1,2,3]. Such materials are used to reduce the oxidation rate of cladding tubes in high-temperature steam by forming protective oxide films such as Cr_2_O_3_, Al_2_O_3_ or SiO_2_, and thus greatly suppress the generation of hydrogen and delay the heating load on the reactor core during severe accidents [4,5]. Two main material approaches to form ATF claddings were proposed. The first one consists of developing brand-new ATF claddings to substitute for Zr alloys, such as SiC_f_/SiC composites [6,7], FeCrAl stainless steels [8,9], Mo alloys [10,11], and MAX phases [12,13]. The second approach, referred to as ATF-coated cladding, is to prepare coatings on the existing Zr alloy claddings, which is the most promising for engineering implementation within the short term. Consequently, many coatings such as Cr and Cr-based alloys [14,15,16], nitride ceramics [17,18,19], FeCrAl alloys [20,21,22] and MAX phase ceramics [23,24,25] are used worldwide. Among them, Cr coatings show resistance to pressurized-water corrosion [26,27], friction and wear [16,28], high-temperature steam oxidation [29,30,31,32] and ion or neutron irradiation [33,34,35,36], and hence become the most promising candidate for advanced ATF cladding coatings. At present, the most important task for the development of pure Cr coatings on Zr alloy claddings is to precisely control the coating to meet the requirements of engineering design.

Engineering feasibility of coating methods should be based on the general considerations as follows: (i) the coating has high compactness with minimal pores and good adhesion with the substrate; (ii) it has little influence on the substrate without significant changes to the grain sizes and chemical composition, introducing no inclusions of second phase or hydrides, etc.; (iii) the coating process is easy to practice, the uniform coating deposition on long-size tubes is feasible, and the coating quality is controllable. Up to date, representative institutions and companies, including the CEA/Framatome/Areva from France, the KAERI from South Korea and UW-Madison/Westing house from the United States, have been committed to the engineering development of Cr-coated Zr alloys tubes. However, their coating technologies are different due to possible issues involved with intellectual property (patents). For example, the French companies mainly adopted high ionization-rate pulsed magnetron sputtering technology (HIPMS) [16,28], the Korean institute chose a 3D laser melting and coating technology (3D-LMC) [37,38], and the Americans applied the cold spraying technology (CS) [39,40]. A detailed description of these coating technologies can be found in the earlier published review papers [41,42]. Meanwhile, the Nuclear Power Institute of China (NPIC) prefers to use the arc ion plating (AIP) technology that also met the coating requirement.

Although it was confirmed that the performances of Cr-coated Zr alloys were improved to some extent as compared to uncoated Zr alloys, differences in the coating methods and parameters were indeed existent and played important roles in their use. For example, grains of CS Cr coatings were presented with large plastic deformation, leading to enhanced irradiation stability as compared to annealed Cr coatings [43]. The mechanical strength of 3D-LMC Cr-coated Zr-4 tubes was higher than that of Cr-coated Zr alloys prepared by other methods due to the formation of martensite structures from a rapid solidification process [38]. Cr coatings prepared by atmospheric plasma spraying (APS) were reported with structural porosity, resulting in inner oxidation of the coatings at high temperatures [44]. Furthermore, a study evaluating the use of AIP and magnetron sputtering (MS) preparation indicated that more holes/cavities were located on the surfaces of MS Cr coatings with cracking evident after air oxidation at 800 °C [45]. Kashkarov et al. [46] prepared a 4.5 μm thick Cr coating with a dense and fine-grain structure by arc-magnetron sputtering and with better resistance to high-temperature steam oxidation than that of comparative samples (6–9 μm thick coatings with coarser columnar crystal) prepared by no-arc magnetron sputtering. Furthermore, the differences in the processing parameters of the same method show great differences in high-temperature oxidation resistance [47,48,49,50]. Our recent studies found that increases in the (110) preferred orientation of the AIP Cr coatings resulted in a decrease in the size and number of cavities during Au^+^ ion irradiation (up to 20 dpa) [34]. Our ongoing research also indicates that different process parameters in the same technology would affect both the resistance to coating cracking in mechanical tests and the degradation behavior of the Cr coatings during high-temperature oxidation.

Up to now, the control of Cr coatings has been performed by studying the effects of deposition parameters on their microstructures, such as surface droplet particles, crystal preferred orientation, grain structures, etc. [47,48,49,50,51,52,53,54]. However, these studies only compared single factor variables and could not distinguish between experimental error and the influence of various parameters. This leads to a lack of systematic guidance for engineering practice. Therefore, facing massive data in microstructures, coating properties and depositing parameters, the orthogonal analysis commonly used as an experiment simplification technique is a good way to deal with the influence of various parameters [55], especially in shifting from small-sized samples to big engineering workpieces in a reliable manner.

In this work, the L9(3^4^) orthogonal tests were used to prepare samples of 1400 mm-long Cr-coated N36 cladding tubes in an industrial multiple arc source system. Large amounts of statistical data based on the orthogonal tests were analyzed to clarify the influence of orthogonal processing parameters on the intrinsic characteristics of coatings, including thickness distribution, deposition rate, surface roughness, defects (droplet particles and pores), and crystal and grain structure. The involved mechanisms are also carefully discussed. This systemic study is believed to be significant for the further development and optimization of full-size Cr-coated Zr alloy claddings by arc ion plating.

## 2. Materials and Methods

### 2.1. Materials

The substrates used were N36 alloy (a novel zirconium alloy developed by NPIC) tube segments (1400 mm) with a nominal composition of Zr-1Nb-1Sn-0.3Fe. These segments were cut from the “full-length” (4000 mm) cladding tubes with an outer diameter of 9.5 mm and a wall thickness of 0.57 mm. The surfaces of these tubes were mechanically polished and the grains were recrystallized. The raw materials for deposition of Cr coatings were Cr targets with a purity ≥99.95% (Shanghai Boyi). Ultra-pure (≥99.999%) Ar (Shanghai Yuanyang) was used as the gas during the deposition process.

Prior to use, N36 tubes underwent multiple ultrasonic cleaning and rinsing using an environmentally friendly agent (Shanghai Bo’er, mainly composed of potassium hydroxide and 2-aminoethanol) and ultra-pure water, respectively. After this, the inner and outer surfaces of tubes were dried with compressed air and kept for storage. Before deposition, the outer surfaces of tubes were repeatedly wiped with dust-free clothes dipped in absolute ethanol (analytically pure CH_3_CH_2_OH, Zhejiang Tengyu) and purged with compressed air to remove the floating dust off the surface.

### 2.2. Coating Preparation

For deposition of Cr coatings on N36 cladding tubes by AIP method, industrial multi-arc ion plating equipment with a vertical side-door configuration, shown in Appendix A), was used. This facility is equipped with 16 asymmetric arc sources arranged in 4 columns (including magnetic systems and cathode targets connected to the arc current power sources). The target–substrate distance is ~200 mm and the effective deposition length is 1400 mm. Cladding tubes were placed in the deposition chamber with two ends inserted into the sleeves on the stations located in the outer ring of a two-stage rotating frame (connected to the bias power source). By this frame, the workpieces could be rotated around their axes, while the frame itself rotated with a constant velocity to ensure circumferential uniformity of deposition.

Appendix A) shows the coating fabrication process. After loading tubes, the deposition chamber was evacuated to a pressure of 5.0 × 10^−3^ Pa. The moisture and air remaining in the chamber walls and cladding tubes were then removed by heating up to the desired temperature. When the pressure was restored back to 5.0 × 10^−3^ Pa, plasma treatment was applied to further clean and activate the surfaces of Zr alloys. This process was carried out at the negative bias voltages of −(700–900) V and a constant gas pressure of 1.2 Pa, respectively.

Cr coatings were deposited in two stages. The initial (priming) stage was carried out to generate a basic layer at negative bias voltages decreasing steply from −400, −300 to −200 V (each step for 5 min), a gas pressure of 0.6 Pa and an arc current of 90 A. In this process, a strong ion bombardment effect induced by high negative bias voltages and low gas pressure can cause the incident ions “pin” into the substrate. In the second stage, the negative bias voltage was reduced to the required value for a smooth transition to the following steady-state deposition, during which the parameters were kept fixed. In this work, the effects of four key parameters during steady-state deposition, namely heating temperature, arc current, negative bias voltage and gas pressure, were investigated.

It should be noted that the heating temperature during coating deposition should not exceed the recrystallization point of the Zr alloys (500–580 °C) to avoid the impact of the coating preparation process on the microstructure and properties of the substrates. Taking into account that arc evaporation and ion bombardment effects during AIP deposition cause a 70–130 °C rise in environment and substrate temperature, the heating temperature should be maintained below 400 °C. The other values were selected empirically based on the process parameters for preparing pure metal coatings by the AIP equipment used. For example, arc discharge could not be ignited successfully with an arc current (<50 A) or a gas pressure (<0.3 Pa) that was too low. Oppositely, an arc current that was too high (>180 A) would lead to an undesirable rise in temperature and an obvious degradation of the coating surface quality; a gas pressure (>2.5 Pa) that was too high would result in the instability of the arc spots, which may damage the cathode components once the arc spots move apart from the range of target materials. Moreover, the bias voltage was applied to improve the coating compactness. However, a bias voltage that is too high (>−200 V) would introduce high growth stress, leading to the peeling of thick coatings during deposition. Based on the above considerations, the samples were prepared at the parameters of L9(3^4^) orthogonal experiments, as shown in Table 1. The deposition time of each orthogonal test was 10 h.

### 2.3. Characterization

The coating thickness was measured from the cross-sectional images of Cr-coated N36 tube samples taken by optical microscopy (OLYMPUS OLS4000). The measurement details are illustrated in Appendix A). The total cross-sectional images were 25 per sample. The mean thickness values were calculated by averaging the measurement results obtained in three random points at each circumferential position.

The three-dimensional (3D) profiles of Cr-coated surfaces and the surface roughnesses were measured by white light interferometer (BRUKER Contour GT-K1). The crystal structures of coatings were characterized by X-ray diffraction (XRD, PANalytical Empyrean). The micro-morphologies of the coating surface and cross-section were observed by scanning electron microscope (SEM, FEI NOVA NanoSEM 400). The sizes and the number of droplet particles on the coating surface were determined from SEM images using the Image J software.

Electron backscatter diffraction (EBSD) was used to obtain the grain structure patterns of the Cr coatings and Zr alloy substrates from polished cross-sectional samples of Cr-coated N36 tubes. The surfaces of cross-sectional samples were treated by using vibrational or ion-beam polishing to eliminate residual surface stress. Before the EBSD examination, a coordinate system for the tube sample was defined using three directions, which are axial (AD), tangential (TD) and radial (RD), respectively (Appendix A). This was undertaken in order to accurately describe the micro-textures of the sample. The grain sizes and micro-textures were evaluated from EBSD patterns by using the HKL Channel 5 software (Oxford Instruments, Abingdon, UK).

### 2.4. Orthogonal Analysis

In the orthogonal analysis, the range and variance of coating indexes were calculated to identify the primary- and secondary-order factors according to their impact on the results as well as to find out the variation of results with these factor values [55]. Those indexes are hereinafter particularly referred to as the coating deposition rate, surface roughness, droplet particle size and number, preferred crystal orientation, etc. The methods of the analyses are introduced in Appendix B, taking the L9(3^4^) orthogonal experiments as an example.

#### 2.4.1. Range Analysis

In the schema of range calculations (Table A1 in Section A.1), the range (*R_j_*) reflects the rangeability of the test index changing with the level of the factor in column *j*. The greater the *R_j_* is, the higher the impact of this factor on the results is and the more important this factor is. The optimal level of the factor in column *j* can be obtained by the size of *k_mj_*.

#### 2.4.2. Variance Analysis

Although the range analysis is simple and clear, it cannot distinguish whether the difference in results is caused by the change in experimental conditions or by errors. It also cannot enable us to understand whether the effect of the factor being considered is significant. The analysis of variance is intended to avoid the above shortcomings [55].

Based on the results of variance calculation (Table A2 in Section A.2.) and F-testing (Table A3 in Section A.2), the factor being analyzed is considered to have a significant impact on the experimental results if *F* > $F_{\alpha }\left(df_{j},{\;df}_{e}^{△}\right)$ and to be insignificant otherwise. The smaller the value of α is, the higher the significance level is.

## 3. Results

### 3.1. Appearance

Appendix A) shows the as-prepared Cr-coated N36 cladding tubes. It can be seen that they are of good straightness. The coating surfaces are evenly colored throughout the whole tube lengths and seem dense without any visible pits, holes or protrusions. An on-site destructive inspection by smashing two samples showed that the Cr coatings firmly adhered to the surfaces of the cladding tubes even in the most severely deformed or cracked parts, indicating their excellent adhesion performance. In Appendix Ab, the appearances of Cr-coated N36 orthogonal samples obtained under nine different process conditions are compared. Obvious differences can be found in the metal luster of the Cr coatings of different samples. It was found by subsequent measurements of coating thickness and surface roughness that a darker color corresponds to a thicker Cr coating or higher roughness which enhances the scattering of light.

### 3.2. Coating Thickness and Deposition Rate

It can be preliminarily concluded from Appendix A that the thickness of the Cr coating is very uniform in the circumferential direction. Statistical relative average deviation (RAD) was used to characterize the uniformity of coating thickness as follows:(1)$$RAD=\sum_{1}^{n}\left|x_{i}-\bar{x}\right|\;/\;n\bar{x},\;n\leq 5$$
where *RAD* is expressed as a percentage (%); $x_{i}$ is the coating thickness measured at each position; $\bar{x}$ is the average coating thickness; and n is the number of measuring points. The smaller the *RAD* value is, the smaller the dispersion of data is, and the better the coating thickness uniformity is.

Table 2 shows the statistical results for only one sample due to space limitations. The data obtained for nine Cr-coated N36 samples provide the circumferential and the axial RADs in the ranges of 1.23–4.24% and 4.29–8.75%, respectively. These results indicate that the circumferential uniformity of the Cr coating thickness is better than the axial one. The former is set by the planetary rotation mechanism of the used AIP equipment. Specifically, during deposition, a cladding tube rotates around the central axis of the vacuum chamber together with the rotating frame. It also turns around its own axis, which ensures the uniformity of circumferential coating on the surface of an N36 tube at a fixed height. On the other hand, coating thickness has relatively poor axial uniformity along the cladding tube, which is thicker in the middle of the tube and thinner at both ends (see Figure 1). Such inhomogeneity arises due to the “shrinkage” of ion trajectory under the action of the electric field [56], which belongs to the inherent characteristics of AIP technology (more details discussed in Section 4.1).

Figure 1 presents the average coating deposition rates at different axial positions of Cr-coated N36 tubes which are calculated according to the statistical data on coating thickness. It can be seen that the deposition rates of the nine orthogonal samples are high in the middle and low at both ends of a tube. Moreover, the uniformity of deposition rate improves moving from the ends to the middle part of the tube as is demonstrated by the flattening of the central parts of the curves presented in Figure 1. These results should be interpreted so that the axial distribution of deposition rate is the inherent characteristic of the AIP equipment and has little correlation with process conditions. One should take into account, however, that the value of the deposition rate itself is related to the parameters of the deposition process.

Figure 2 shows the thickness of the Cr coating for a representative sample (#1) versus the deposition time at different axial positions. The results of the orthogonal experiment carried out under different process conditions are similar. Linear dependence of the thickness of the Cr coating on the deposition time indicates the stability of the deposition rate in the investigated time range under the applied process conditions. Such stability facilitates a simple control of coating thickness. It can be also seen that the slopes of the three curves measured at different axial positions are different. The slope of the curves increases which means the growth of the deposition rate upon the increase in the distance from the tube end (close to the central tube part in the axial direction). This is consistent with the deposition rate distribution shown in Figure 1. Furthermore, the rise in deposition time leads to an increase in the difference between the coating thicknesses at the end and in the middle part of the cladding tube. This means that the thicker the coating is, the lower the thickness uniformity is.

Table 3 and Table 4 show the results of the orthogonal calculations and the F-testing based on the average deposition rate. According to the calculated *R_j_* values, the primary- and secondary-order factors are arc current > negative bias voltage, heating temperature > gas pressure and the influence of gas pressure is expected to be negligible due to a very low *SS_j_* value obtained by variance calculations. Therefore, in the subsequent F-testing, the influence of gas pressure is considered an experimental error. The F-testing results show that the significance level of the influence of arc current on the deposition rate of the Cr coatings is about 20 times higher than those of heating temperature and negative bias voltage.

Appendix A) presents the average deposition rates of the Cr coatings versus the arc current, negative bias voltage and heating temperature. The mechanisms of these variables influencing the coating deposition rate will be discussed in detail in Section 4.2. It should be noted that, when compared to the influence of arc current, the change of deposition rate caused by the other two factors is much smaller, which is consistent with the results of F-testing. Referring to Figure 1, one can also see that the deposition rates of the orthogonal samples are divided into three groups by the value of the arc current: 150 A (samples # 1, 4 and 7), 120 A (samples # 2, 5 and 8) and 90 A (samples # 3, 6 and 9). Among them, the samples # 4 (300 °C, 150 A, −80 V) and 9 (250 °C, 90 A, −160 V) have the highest and lowest deposition rates, respectively, which correspond to the most (highlighted in red bold in Table 3) and the least optimum combination of heating temperature, arc current and negative bias voltage values.

### 3.3. Micro-Morphology and Roughness of Coating Surface

Figure 3 shows the SEM images and 3D contours of the surface morphologies of the Cr coatings. Appendix A) shows the profiles of surface droplet particles obtained via Image J software at a lower magnification. It can be seen from these images that many droplet particles with different sizes are homogeneously distributed on the whole coating surfaces of nine orthogonal samples, which is also reflected in the 3D surface profiles. Table 5 presents the results of the statistical analysis of surface droplet particles obtained by the Image J software for different samples.

Most of the particles have small diameters of 0.5–1 μm, while larger particles appear occasionally. The particle statistics are quite different for different samples. Much more particles and larger particle sizes are found on the surfaces of samples # 1, 4 and 7 as compared to those of other samples, indicating that certain process parameters have a greater influence on the droplet particle formation. Furthermore, a comparative analysis shows that the values of surface roughness, Ra, provided by the insets in the panels of Figure 3, strongly depend on the number and size of surface particles.

In order to understand the influence of process parameters on the surface roughness of coating and particle formation, orthogonal analyses using the Ra values and particle numbers were carried out. Appendix A (in Appendix A) list the results of range and variance calculations and F-testings, respectively. The former results show that the primary- and secondary-order factors influencing the surface roughness are: arc current > heating temperature, negative bias voltage > gas pressure, while for the number of particles these factors are: arc current > heating temperature, gas pressure > negative bias voltage. In the F-testings of surface roughness and particle numbers, the effects of gas pressure and negative bias voltage are considered within the experimental errors because of their very small *SS_j_* values. The results of F-testings show that the arc current has a rather significant effect on both the surface roughness and particle number, especially for the latter. Appendix A (in Appendix A) selectively reports the significant correlations based on the F-testing results presented in Appendix A (i.e., roughness—arc current, roughness—bias voltage, roughness—temperature, particle number—arc current). The influences of these variables on the surface roughness and droplet particles will be detailed in Section 4.3.

Moreover, the orthogonal analysis of particle sizes was carried out showing the following order of factor influences: arc current > negative bias voltage > gas pressure > heating temperature. The particle size is directly proportional to the arc current and inversely proportional to the negative bias voltage. It decreases first and then increases with the increase in gas pressure. However, the significance levels of all these factors are relatively low. The highest one of arc current corresponds to α ~ 0.11 (nearly 40 times compared with that of the particle number). The respective results conditioned by these factors are therefore not shown here.

It should be noted from Appendix Aa,d that the dependences of surface roughness and the number of particles on the arc current are very similar, indicating that the number of particles on the coating surface is the main source of surface roughness. The much higher surface roughness and numbers of particles for samples # 1, 4 and 7 shown in Figure 3 and Table 5, as compared to other samples, are related to the largest arc current value of 150 A. The second highest values of the discussed characteristics of coatings are observed for the samples # 2 (350 °C, 120 A, 120 V), 3 (350 °C, 90 A, −80 V) and 5 (300 °C, 120 A, −160 V). The number of particles on the surface of sample # 3 is the least among the three latter samples due to an arc current value of only 90 A. However, the surface roughness for this sample shows no significant difference from that of sample # 2, which may be explained by larger particle sizes for the former sample (Table 5). This means that the surface roughness of the coating is determined not only by the number of particles but may also be affected by particle size. Our results are consistent with the ones obtained by Gong [57] in the study of TiN films.

### 3.4. Micro-Morphology of Coating Cross-Sections and Internal Defects

Figure 4 shows cross-sectional SEM images of the Cr-coated N36 orthogonal samples taken in the backscattered electron (BSE) mode. It can be seen that the thicknesses of the Cr coatings obtained under different process conditions are different. At a constant deposition time, the coating thickness is determined by the rate of deposition. As stated above, it mainly increases with the increase in arc current value. In Figure 4, one can see a distinct contrast between the Cr coating and the Zr alloy substrate and a very sharp, clear, flat and uniform interface. Except for a few pores found at the Cr/Zr interfaces (noted by red circles) of samples #1 (Figure 4a), 4 (Figure 4d) and 5 (Figure 4e), no interfacial defects such as pores or cracks are observed for other samples, indicating that the as-prepared Cr coatings have good adhesion to the N36 substrates. The upper edges of the coatings also look very smooth and flat, and the surface particles shown in Figure 3 are not observed. This may be related to the detachment of these particles during metallographic sample preparation.

In addition, color contrast among different grains of Cr coatings is visible in the SEM-BSE images, which may be related to the difference in grain orientations. This contrast weakens with a decrease in coating thickness, which may be related to the decrease in grain sizes. For sample # 9 (Figure 4i), the grain boundaries become blurred because of too small grain sizes. It was also found that at a high arc current (150 A), many interior pores appear in the Cr coatings (indicated by white arrows), as can be seen in the cases of samples # 1 (Figure 4a), 4 (Figure 4d) and 7 (Figure 4g). Nearly all the pores have arcuate shapes and are located below the droplet particles embedded into the coatings, indicating that their formation is caused by a typical shadowing effect [58]. On the contrary, no obvious such pores are found in the other orthogonal samples, especially in the Cr coatings prepared at a low arc current (90 A).

### 3.5. Crystal Structure and Preferred Orientation of Cr Coatings

Figure 5 presents the XRD results of nine Cr-coated N36 orthogonal samples. In comparison with the standard PDF card, the surface coatings of all samples are found to be pure phases of Cr metal with negligible impurity phases present. The three primary diffraction peaks are clearly defined and located at 44.4°, 64.6° and 81.7°, respectively. These correspond to the (110), (200) and (211) planes of a Cr crystal with a body-centered cubic (BCC) structure, indicating a high degree of crystallinity in the Cr coating. The interplanar distances are calculated to be 2.0382, 1.4417 and 1.1773 nm, respectively, which are consistent with the PDF card values. The diffraction peak of the (110) plane is much stronger than the other two planes, consistent with results obtained during other studies [27,48,49,59,60]. This is the close-packed plane of a BCC crystal, which will have the lowest surface energy and smallest growth energy barrier. In addition, no obvious drift or asymmetric changes were found in either the position or shapes of the three diffraction peaks, indicating that the internal stress and lattice distortion in the Cr coatings are small. The full width at half maximum (FWHM) of the three primary peaks was measured to increase gradually from (110) to (211), with this result implying improved crystallinity of the growth along the low-index planes. Nevertheless, the diffraction peak intensities of the various Cr-coated N36 samples were different, meaning that differences may exist in the crystal, including orientation.

The texture coefficient (TC) of each sample was calculated by the following formula [61]:(2)$$TC_{hkl}=n(I_{hkl}/I_{hkl}^{0})/\sum^{\;}(I_{hkl}/I_{hkl}^{0})$$
where $I_{hkl}$ is the measured intensity (a.u.) of the (*hkl*) plane diffraction peak; $I_{hkl}^{0}$ is the relative intensity (%) in the standard PDF card; n is the number of planes considered, which here is *n* = 3. If $TC_{hkl}$ is greater than 1, the (*hkl*) plane has a preferred orientation, with higher values indicating a greater extent of preferred orientation [61,62]. The results from these calculations are summarized in Table 6.

The calculated results demonstrate that the TC_110_ of the samples were all greater than 1, indicating that the preferred orientation of the (110) plane of the Cr coatings is a common phenomenon. This result is consistent with the principle of minimum energy during crystal growth. However, there are obvious changes found among the *TC* values of the three planes of each different sample. For example, the calculated TC_110_ values for samples # 3, 5, 7 and 9 decreased significantly, while TC_200_ and TC_211_ increased. A preferred orientation of (211) and (200) planes is evident in samples # 5 and 9, respectively (highlighted in red bold in Table 6). The analysis would infer that the process parameters are able to alter the crystal growth orientation of the Cr coatings, which was also confirmed in other studies [48,49,63].

To determine the relationship between the differences in the calculated $TC_{hkl}$ values and process parameters, orthogonal analyses were performed with TC_200_ and TC_211_ as indexes, with the results summarized in Appendix A (in Appendix A).

In Appendix A), the F-testing results report that the significance level of various process parameters on the preferred growth behavior of the (200) plane is much lower than that of the (211) plane. This is potentially because the (200) plane is the second low-energy plane. Although less common in comparison to a preferred orientation of the (110) plane, the “adaptability” of the (200) plane to the growth environment is still higher than that of the (211) plane, which has the highest energy barrier among the three. In this scenario, the (200) plane is considered as equally insensitive to process parameters as the (110) plane. Based on this result, we will mainly examine the influence of process parameters on the preferred growth behavior of the (211) plane.

Analysis of the (211) plane shows that the gas pressure had the greatest impact, followed by the negative bias voltage and arc current. Over the range of parameters investigated during this study, the temperature contributed little to the growth of this high-index plane (Appendix A) and can be considered an experimental error. This may be due to the temperature rise (70–130 °C) caused by arc evaporation and the ion bombardment effect of AIP being close to or exceeding the variation in temperature set values, potentially masking the impact of temperature on crystal growth.

Appendix A) shows the variation in the degree of preferred orientation of the (211) plane as a function of gas pressure, negative bias voltage and arc current, respectively. A detailed demonstration of the effects of these variables on the TC_211_ can be referred to in Section 4.5. One should note that the process parameters of sample # 5 (120 A, 0.8 Pa, 160 V) are all at optimal values (k_mj_ are highlighted in red bold in Appendix A), with the TC_211_ value (>1) being the largest (Table 6).

### 3.6. Grain Structure of the Cr Coatings

Figure 6 exhibits the EBSD characterization results of the grain structures of the Cr-coated N36 alloy orthogonal samples. Figure 7 presents the corresponding positive and reverse polar diagrams of the Cr coatings. EBSD analysis clearly demonstrates the presence of either fibrous or columnar grains in the Cr coatings with a trend for growing upward perpendicular to the Zr alloy substrate. The grains typically have a “V”-shaped morphology of different lengths, with the larger side located towards the surface. Unexpectedly, the majority of the Cr grains do not comprise the thickness of the coating but are stacked in multiple layers. A layer of fine grains was formed near the Cr/Zr interface, which is expected to be generated during the initial stage of coating preparation. As the parameters utilized during the priming process are consistent, the thicknesses of the fine grain layers in different samples are generally equivalent, with a thickness of approximately 1 μm. The polar diagrams demonstrate the micro-textures of the coating grains (Figure 7). Well defined {100}<001> cubic textures are found in samples # 2, 3, 6 and 8. The {100} planes of Cr with a BCC structure are parallel to the AD-TD plane and the <001> directions are parallel to the RD direction and perpendicular to the coating surface. This is consistent with previous results reported by several studies [30,51,64,65]. However, the textures of samples # 1, 4, 5, 7 and 9 appeared to be weakened, even with some Cr grains tending to be randomly oriented. This observation seems to be related to an interruption of the growth of coating grains by the large arc current (150 A) or negative bias voltage (−160 V).

Table 7 reports the equivalent average diameters ($\bar{d}$) and aspect ratios ($\bar{L/W}$) of the Cr grains determined by the Channel 5 software. The data of the different samples show that the average diameters of the Cr grains are 1–2 μm, with measured maximum values of 3–12 μm. The average aspect ratios are 3–4, with maximum values of 12–23. The statistical results of # 8 and 9 deviate from the overall set averages, showing significantly different maximum and minimum values (highlighted in red bold in Table 7). Referring to the EBSD patterns in Figure 6, one can see that sample # 8 has a reduced number of coarse-sized grains, with clearly defined columnar morphologies, whilst sample # 9 has a larger number of fine grains, which generally are shaped as narrow and short pins.

To understand the influence of various process parameters on the coating grain structure, the equivalent average diameter and aspect ratio of the grains are used as indexes for orthogonal analysis. Appendix A report the results of range and variance calculations and F-testings. The former results show that the primary- and secondary-order factors influencing the grain size are: arc current > negative bias voltage > gas pressure > heating temperature, while for the aspect ratio of the grains these factors are: negative bias voltage > heating temperature > arc current > gas pressure. The significances of heating temperature and gas pressure are low enough (very small *SS_j_* values) to consider the impact of the two factors as falling within the experimental errors during the F-testing. The significance levels show that the sensitivity of the grain size to the process parameters is lower than that of the aspect ratio (Appendix A). One possible explanation is a more significant influence of coating thickness on the grain size than that on the aspect ratio, reducing the response of the former to process parameters.

Appendix A presents the variation of the average grain size and aspect ratio of the Cr grains with respect to various process parameters. Section 4.6 elaborates on the influences of these parameters on the coating grain structure. In particular, a synchronous trend for the grain size and aspect ratio can be noted in Appendix Aa–d, implying that columnar crystal growth is an inherent feature of AIP-deposited Cr coatings. Moreover, the variation of the above two indexes with respect to the process parameters (excluding arc current) is extremely similar to that of the coating deposition rate (Appendix A). Consequently, the average size and aspect ratio of the Cr grains are also related to the rate of coating deposition or thickness. At least, the size of grains will grow with the thickening of the coating and columnar grains will become either wider or longer under the same process condition.

## 4. Discussion

### 4.1. Axial Distribution of Coating Thickness

From the point of view of the physical processes, the axial distribution of coating thickness largely depends on the number of material ions incident at a certain position on the surface of the cladding tube in the axial direction. As we know, positively charged target material ions moving in the vicinity of the substrate are accelerated by the space sheath and bias fields. However, charged particles also deflect under the action of an electric field. Let us assume that a charged particle moves between two electrode plates perpendicular to the direction of the electric field. The particle is offset in the direction of the electric field when it leaves the field region, and can be easily calculated according to Coulomb’s law and Newton’s law of motion as follows:(3)$$y=QUL^{2}/2mdv_{0}^{2}$$
where Q is the particle charge (C); U is the voltage between two electrode plates (V); L is the plate length (cm); m is the mass of the charged particle (kg); d is the plate spacing (cm); and $v_{0}$ is the initial incidence velocity (m/s) of the charged particle. Therefore, as long as the electric field distribution is known, the trajectories of charged particles in the electric field can be found.

Zhao [56] simulated motion trails of Cr ions in the bias field in a cylindrical deposition chamber. A shrinkage phenomenon for the motion tracks of Cr ions under the action of a bias field was found and called “ion shrinkage” by the author. The results of these calculations indicate a motion trend of Cr ions towards the center of the substrate, which would lead to higher coating thickness at the center of the substrate as compared to that at its ends or rims. Figure 8 shows the distribution of the thickness of the Cr coating on the surface of the cladding tube obtained by evaporating a single Cr target with a diameter of 156 mm at a target–substrate distance of ~200 mm. The coating thickness distribution within the effective deposition range (~250 mm) of a single arc source is close to a Gaussian distribution, which is a good demonstration of the above inference. On the other hand, Cr ions evaporated from the targets can have multiple charge states (Cr^+^, Cr^2+^ and Cr^3+^). Therefore, the offsets of Cr ions with different charges in the bias field are also different [56], which enhances the non-uniformity of coating thickness to a certain extent.

The calculated results reported by Zhao also present a more obvious shrinkage trend for the Cr ions with a stronger electric field intensity [56]. However, for our Cr-coated N36 samples, no significant influence of negative bias voltage on the axial distribution of Cr coating thickness along the tube was found (Figure 1). This may be related to the spatial arrangement of arc sources in the used AIP equipment. For a single arc source, the ion shrinkage effect described above may be easily observed. However, its influence can be attenuated when the effective deposition ranges of multiple arc sources overlap and the substrate is placed within them. As described in Section 2.2, the equipment used in our experiments comprises 16 asymmetric arc sources arranged in four columns with overlapping effective deposition ranges. The top and bottom sources are intended for compensation purposes, which improves the uniformity of the Cr coating in the axial direction of the cladding tube. The results presented above show that the influence of the configuration of equipment on coating uniformity is much more significant than that of process parameters.

### 4.2. Effect of Process Parameters on Deposition Rate

The coating deposition rate is directly proportional to the net ion flux received by the substrate, which is determined by the number of ions incident and escaping from the substrate per unit of time. As depicted in Appendix A, the deposition rate of the Cr coating monotonically increases with the increase in arc current. It can be easily understood that the increase in arc current leads to higher numbers of cathode arc spots and more intensive evaporation and ionization of target materials [66]. Such processes result in a higher plasma density and a larger flux of Cr ions onto the substrate thus increasing the deposition rate.

The second significant factor is the negative bias voltage, the increase of which reduces the deposition rate (Appendix A). The function of the negative bias voltage is to accelerate Cr ions by the bias field, which brings about two competitive effects. On the one hand, when the arc current remains unchanged, the increase in negative bias will increase the number of ions reaching the substrate per unit of time, which enhances the coating deposition. On the other hand, a strong bias field endows ions with high energy. These high-energy ions can induce a bombardment effect on the substrate. In particular, the energy transmitted to the substrate heats it, which facilitates lateral migration of adsorbed atoms on the substrate or coating surface and makes the coating more compact [67]; however, this reduces the deposition rate in the direction perpendicular to the substrate. Moreover, if the energy exceeds the sputtering threshold of coating atoms, the adsorbed atoms will be sputtered off the coating surface also reducing the deposition rate. It is obvious that for the parameters used in our study, the ion bombardment effect caused by negative bias voltage is more significant than the effect of enhanced ion deposition.

As compared to the first two factors, the effect of heating temperature on the deposition rate is relatively weak. Appendix A shows that the coating deposition rate first increases and then decreases with the increase in temperature. Since the plasma formed by arc discharge during deposition is quasi-neutral [68], it is reasonable to assume that the expression for the average free path of ideal-gas molecules may be applicable:(4)$$\lambda =k_{B}T/\sqrt{2}\pi d^{2}P$$

Here, T is the temperature (K); d is the effective particle diameter (m); P is the gas pressure (atm); and $k_{B}$ is the Boltzmann’s constant ($k_{B}=1.38\times 10^{-23}$ J/K). According to Equation (4), under a certain pressure, the increase in temperature leads to an increase in the average free path in plasma and a decrease in the mutual collision frequency of particles. These effects lead to a higher flux of target material ions reaching the substrate and thus the deposition rate becomes higher. However, upon the rise in temperature, the flight distances of particles also increase. Actually, due to the limited volume of the vacuum chamber, ever more particles will reach the chamber walls and other internal components of equipment and deposit on their surfaces. In this way, a significant portion of the material is lost instead of being deposited onto the substrate.

### 4.3. Effect of Process Parameters on Droplet Particles and Surface Roughness

In the process of AIP deposition, materials violently evaporate due to the high current density (up to 10^10^ A/m^2^) at the arc spots on the cathode target surfaces [66]. After this, the steam flow carries splashed droplets away [69,70]. The arc current has the most significant effect on the formation of droplet particles on the coating surface because it is a direct energy source for droplet emission. Generally, more energy is supplied to the cathode targets with the increase in arc current, resulting in an increase in the number of arc spots and the total volume of molten pools [66]. Consequently, the evaporation of target materials is intensified leading to an increase in the number and size of emitted droplets. The droplet emission can be estimated by the following formula [71]:(5)f(r)=Qexp(−ar)
where Q is the total quantity of electricity of the cathodes which is proportional to arc current; a is the correlation constant of cathode material; and r is the average radius of droplets, respectively. According to Equation (5), the number of emitted droplets with the same radius or the droplet radius at a constant number of droplets increases with the increase in arc current, thereby the coating surface roughness also increases (Appendix Aa,d).

The influence of heating temperature and a negative bias voltage is less significant because these parameters affect only the droplet motion. A rise in temperature enhances the kinetic energy of droplets and simultaneously leads to an increase in the average free path of gas, reducing the collisions between Ar atoms or ions and droplets. This results in an increase in the number of particles deposited on the coating surface per unit of time. Therefore, the coating surface becomes rougher when the heating temperature increases (Appendix A). The effects of negative bias voltage on the coating surface roughness are more complicated as compared to the temperature ones. The results of the orthogonal analysis show that the roughness of the coating surface decreases first and then increases with the increase in negative bias voltage (Appendix A). At this, the number of particles demonstrates an opposite trend and the particle sizes monotonically decrease (Section 3.3).

We explain the observed phenomena by the ion bombardment effect and electric repulsion caused by the negative bias voltage. With the increase in voltage, the ions are endowed with more energy and the ion bombardment effect enhances. The macro-particles on the coating surface are possibly broken by high-energy ions and split into smaller ones, which can result in a decrease in roughness if dominated by particle size. On the other hand, during the motion of droplets in the plasma, electrons migrate faster than positive ions and are more likely to accumulate on the droplet surfaces, making them negatively charged [72]. The negatively charged droplets getting into the space sheath close to the substrate are repulsed by the bias field, thus reducing the number of particles deposited on the coating surface [73]. This also contributes to reducing surface roughness. However, when the negative bias voltage is very high, the ion bombardment may cause intensive sputtering of surface particles, reducing their number. Even so, some particles embedded in the coating can be also broken and sputtered off, which leads to the formation of many pits on the coating surface (as indicated by arrows in Figure 9) and an increase in roughness.

### 4.4. Formation Mechanism of Arcuate Pores

As mentioned in Section 3.4, the arcuate pores in Cr coatings are formed by the shadowing effect. In order to further understand the mechanism, we must consider whether the droplet particle solidified before or after reaching the coating surface. Despite the melting point of Cr being up to 1907 °C, it is still possible that the emitted molten droplets remain in a liquid state before reaching the coating surface due to their rapid flight speed during the deposition process. If so, large deformation of the liquid droplets is sure to happen due to their collision and wetting action with the coating surface. In such a case, the arc-shaped ends of the droplet particles cannot be maintained and arcuate pores cannot be generated in the coating. The study of Keidar et al. [74] obtained the simulated dynamics of droplet temperature in typical equipment with a characteristic length of 500 mm. They concluded that the initial temperature of droplets with characteristic velocities of 10–100 m/s decreases below the melting point after about 10^−3^–10^−2^ s. It can be assumed therefore that the droplets ejected from the molten target material rapidly shrink into quasi-spheres due to surface tension and subsequently solidify during their flight to the substrate surface. However, these solid particles can adhere to the deposited coating surface only with weak van der Waals force. It is clear that most of them will detach from the surface under continuous bombardment by high-energy ions. A small part of them may survive by mechanical occlusion with the coating surface. The incident Cr ions will rapidly agglomerate around the survival particles, causing a shadowing effect [58] on the following incident ions and leaving pores underneath the particles (Figure 10a). Subsequently, the particles themselves will attract more Cr ions, which will be deposited on their tops and gradually develop into columnar grains (Figure 10b).

It should be emphasized that only at a large arc current when many droplets are generated, can a small part of solidified particles be buried in the coating and lead to the arcuate pores. In contrast, almost no pores form in the Cr coatings prepared with a small arc current. This confirms the conjecture that the survival probability of particles on the coating surface is very low during continuous high-energy ion bombardment. The droplet particles observed on the coating surface in Figure 3 are those that are left at the end of the deposition when there is no more high-energy ion bombardment. This finding is very important because the embedded particles reduce the compactness and continuity of coatings, and thus their corrosion and high-temperature oxidation resistance.

### 4.5. Effect of Process Parameters on Crystallographic Orientation

As is well known, the growth of high-index planes requires more external energy in comparison to low-index planes. It is not difficult to infer in terms of energy that with the decrease in the gas pressure, collisions between the Cr ions and Ar atoms/ions will also decrease, thus reducing energy dissipation. This will result in a greater average remaining energy in particles after reaching the substrate surface. The energy is enough for the adsorbed Cr atoms to overcome the barrier of the (211) plane, and to promote their diffusion, migration and lattice arrangement on the substrate surface [63], allowing growth of those planes (Appendix A).

Similarly, with an increase in the arc current, more particles of the target material will be emitted. As the number of Ar atoms/ions at a given gas pressure will remain unchanged, more Cr ions/atoms are able to move to the substrate surface while undergoing fewer collisions, resulting in increased particle energy. However, if the arc current continues to increase when the emitted particles of the target material reach a certain number, the probability of collision and quantity of energy dissipated will also increase. In addition, an increase in the arc current will increase the number of droplets emitted. As the droplet particles may be polycrystalline and have random orientations, this will reduce the preferred orientation of the coating crystals (Appendix A).

An increase in negative bias voltage will increase the kinetic energy of the positively charged ions due to the acceleration effect of the bias field on them. However, within the range of parameters studied in this work, the degree of the (211) preferred orientation does not increase monotonically with the increase in bias voltage (Appendix A), which is inconsistent with the above-mentioned energy barrier-based mechanism of crystal growth. Potentially, an anti-sputtering effect will be produced when the incident ion energy exceeds the threshold energy for sputtering for the coating atoms. At this time, the atoms of the (211) plane that have high surface energy may be the first to escape from the crystal surface, inhibiting the crystal growth along the plane. As the escaping atoms of the (211) plane are energetically saturated, when the negative bias voltage is further increased the sputtering yield of the plane will not increase while the atoms of the (200) and (110) planes with lower surface energy begin to escape. However, as high-energy ions are more conducive to crystal growth along the (211) plane, the preferred orientation will recover once the other two planes are inhibited.

### 4.6. Effect of Process Parameters on Grain Structure

Mattox et al. [75] and Messier et al. [76] proposed that strong ion bombardment can create nucleation sites for incident atoms on the coating. According to the theory of film nucleation and growth, the nucleation rate (I) of the film can be described by the following formula [77]:(6)$$I=Z\cdot n_{1}2\pi r^{\prime }\cdot \sin\theta \cdot J\cdot a_{0}\exp\left[\left(E_{d}-E_{D}-∆G^{*}\right)/kT\right]$$
where Z is the Zeldovich correction factor (~10^−2^); $n_{1}$ is the adsorbed atomic density; $r^{\prime }$ is the critical nuclear radius; θ is the wetting angle between the nucleus and the substrate surface; J is the flux of ions incident on the substrate surface; $a_{0}$ is the distance between adjacent adsorption sites; $E_{d}$ is the surface-atom adsorption energy; $E_{D}$ is the surface diffusion energy of adatoms; $∆G^{*}$ is the change of total free energy during the formation of the critical nucleus; T is the absolute temperature; and k is the Boltzmann constant. If it is assumed that adsorption sites on the substrate surface have equivalent adsorption energies, the nucleation rate is simplified to be related to $n_{1}$, J, $∆G^{*}$ and T for the given substrate and coating materials.

Due to the high negative bias voltage (up to 400 V) used during the priming stage, the high energy of the incident ions endowed by the bias field will increase the adsorbed atomic density ($n_{1}$), with strong ion bombardment leading to an increased rate of nucleation (I) of the Cr coating. The theory of crystal solidification describes that an increase in the nucleation rate will always inhibit grain growth [78]. Practically, this leads to grains in the basic layer having a very small size. In the stage of steady-state deposition, as the negative bias voltage decreases (maximum to 160 V), the ion bombardment will decrease accordingly. This will lead to a decrease in the crystal nucleation rate and an increase in the growth rate, resulting in grains growing with slender and fibrous structures, before gradually developing into coarse columnar structures as transverse diffusion of adatoms occurs. Columnar grains are often featured by narrower bottoms and wider tops, which likely results from a slower lateral growth rate in comparison to axial growth. However, EBSD results (Figure 6) clearly demonstrate that not all columnar grains will develop smoothly throughout the entire coating thickness. The grain structures of the sample Cr coatings under different process conditions will present varying degrees of multi-layer grain stacking. This is most likely due to an ion bombardment process, which is related to the energy of the ions and will periodically result in nucleation during coating growth, interrupting the continuous growth of the fibrous/columnar grains. Therefore, the greater the negative bias voltage, the higher the nucleation rate, the smaller the average grain size and the smaller the aspect ratio (Appendix Ac,d).

The average grain size and aspect ratio are seen to first rise and then fall with increases in the arc current (Appendix Aa,b). This is possibly due to the increase in arc current significantly increasing the temperature (70–130 °C). According to Equation (6), the rise in the temperature will decrease the nucleation rate of the Cr coating by decreasing the undercooling degree of adatoms in the process of condensation and increasing the possibility of desorption of adatoms. Under these conditions, the growth of longitudinal Cr grains is promoted to develop into fibrous/columnar crystals, leading to a simultaneous increase in the average grain size and aspect ratio. However, as the arc current increases further, the flux of ions incident on the substrate surface (*J*) will increase significantly. This will result in the nucleation rate increasing and average grain size decreasing. In the meantime, an increase in the deposition of droplet particles will lead to a disturbance in the continuous growth of the original grains and a reduction in the aspect ratio.

The average grain size was measured to first decrease and then increase as the gas pressure was elevated (Appendix A). This can be explained according to Townsend’s discharge theory. In the process of gas ionization, the increase in gas pressure will increase the emissivity of secondary electrons, leading to increased plasma density and electric field intensity [79]. This will increase the incident energy of ions and the adsorbed atomic density ($n_{1}$), resulting in the nucleation rate increasing and average grain size decreasing. However, as the gas pressure increases further, the collision between Ar and Cr ions will be enhanced. There will also be a decrease in the ion bombardment effect, reducing the nucleation rate and increasing the average grain size.

In addition, the grain aspect ratio will at first increase, and then it will decrease with an increase in heating temperature (Appendix A). As previously discussed, a rising temperature will decrease the nucleation rate of the Cr coating by prompting the desorption of adatoms. The growth of columnar grain will then be promoted, increasing the aspect ratio. As the temperature increases further, the average free path of the gas increases, reducing the collision of atoms/ions and subsequently increasing ion bombardment upon the substrate. This will enhance the nucleation rate, with the promotion of transverse diffusion of adatoms retarding the development of fibrous/columnar grains, which will result in a decrease in the grain aspect ratio.

To summarize, the influence of various parameters on the coating grain structure is complex with many competing influences. The key variable may change with variations in the process conditions. However, we can at least clarify that the key to controlling the grain structure of the Cr coating is changing the crystal nucleation rate. The generally accepted parameter is a negative bias voltage with the simplest influencing effect, the increase of which can create more nucleate sites to inhibit the grain growth to coarse columnar crystals.

## 5. Conclusions

In this paper, orthogonal experiments were used to study the effects of the AIP parameters on the preparation of Cr coatings on the long-size N36 cladding tubes and to discuss the underlying mechanisms. The main conclusions are as follows:(1)The axial distribution of the coating thickness decreasing from the central part to both ends of the cladding tube is attributed to the “ion shrinkage” phenomenon induced by a bias field. This coating uniformity can be improved by overlapping the effective deposition zones of arc sources but is seldom carried out by the processing parameters.(2)The deposition rate of the Cr coating is mainly dependent on the arc current, the increase of which will lead to a monotonic increase in the coating deposition rate by increasing the incident ion flux.(3)The number and size of droplet particles are greatly influenced by the arc current. Particularly, decreasing the arc current can effectively inhibit the droplet emission of the arc source, thus reducing the surface roughness and also improving the compactness of the Cr coatings with less formation of pores induced by droplet particles embedded in the coatings.(4)The change of process parameters will significantly affect the growth of the high-index plane of (211) rather than that of the low-index plane of (110), even though the preferred orientation of the latter is prevailing in the Cr coatings. An enhancement of the growth of the (211) plane can be achieved via a decrease in the gas pressure that reduces the collision between ions and their energy depletion.(5)The key to controlling the grain structure of the Cr coating is changing the crystal nucleation rate. The simplest way is to increase the negative bias voltage to create more nucleate sites, inhibiting the grain growth to coarse columnar crystals. Moreover, coating thickening contributes to the columnar growth of grains.

This article aims to give some methods to control the quality and microstructures of Cr coatings. However, the optimal process or coating thickness cannot be obtained directly from processes # 1–9, but from the combination of optimal levels of various factors. This should be based on further studies on the service performances of Cr coatings, such as mechanics, high-temperature oxidation and irradiation, which is a huge and systematic work.

## Figures and Tables

**Figure 1 materials-15-07177-f001:**
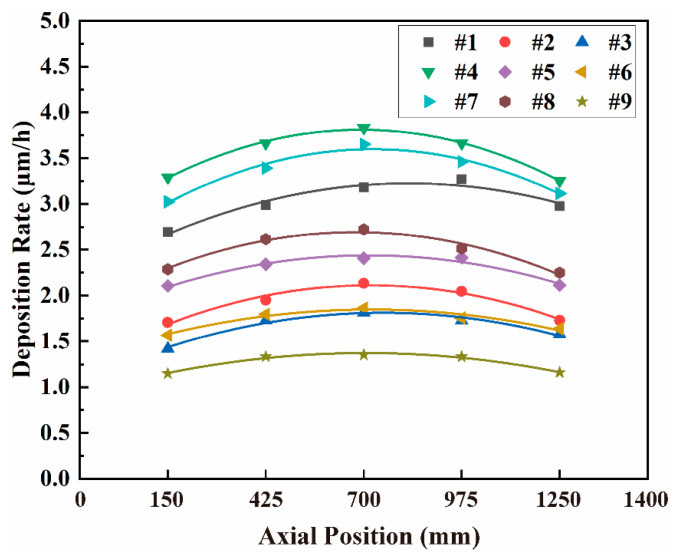
Axial distribution of coating average deposition rate.

**Figure 2 materials-15-07177-f002:**
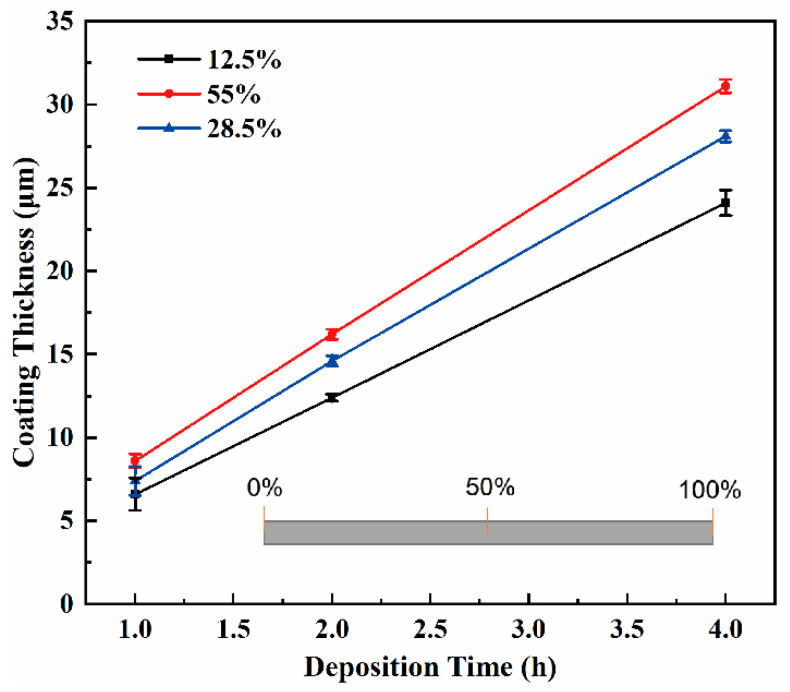
Cr coating thickness versus deposition time at different axial positions of a representative coated tube (sample #1). The percentage values correspond to the ratios of the distance of an axial position to the same end of the cladding tube to the total tube length, as indicated in the inset.

**Figure 3 materials-15-07177-f003:**
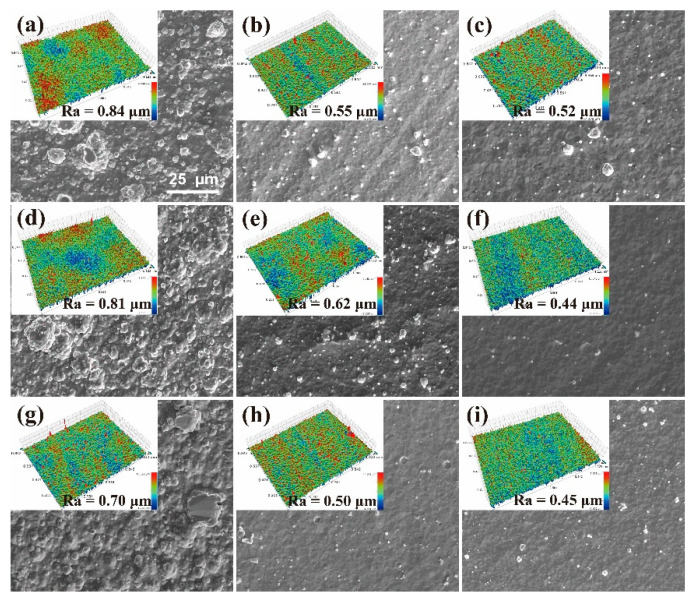
SEM images of the surface morphology of Cr coating: (**a**–**i**) correspond to samples # 1–9, respectively. The upper-left insets show the 3D surface profiles obtained by white light interferometry and provide the values of roughness Ra. All images have the same scale.

**Figure 4 materials-15-07177-f004:**
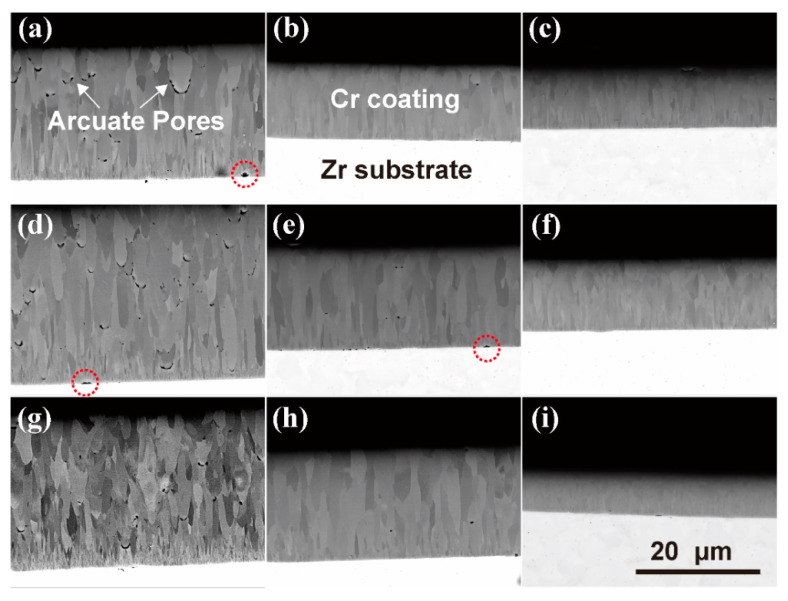
Cross-section SEM-BSE images of Cr-coated Zr alloy: (**a**–**i**) correspond to samples #1–9, respectively. All images are of the same scale.

**Figure 5 materials-15-07177-f005:**
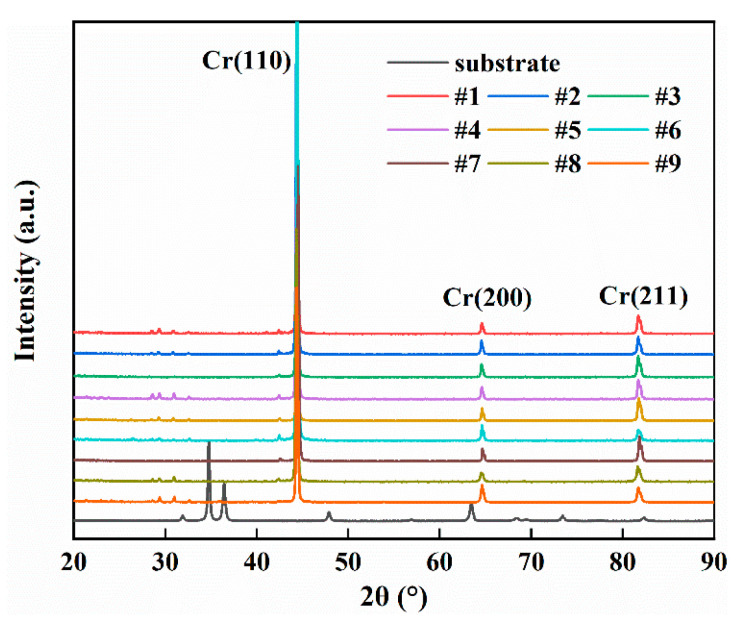
XRD patterns of orthogonal samples.

**Figure 6 materials-15-07177-f006:**
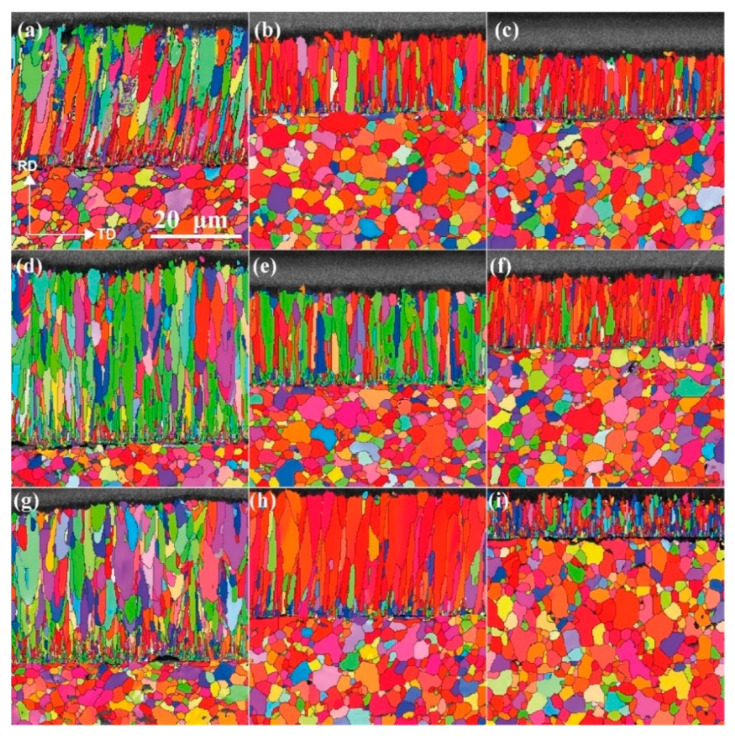
EBSD images of sample grain structure: (**a**–**i**) corresponds to samples # 1–9, respectively. All images are at the same magnification.

**Figure 7 materials-15-07177-f007:**
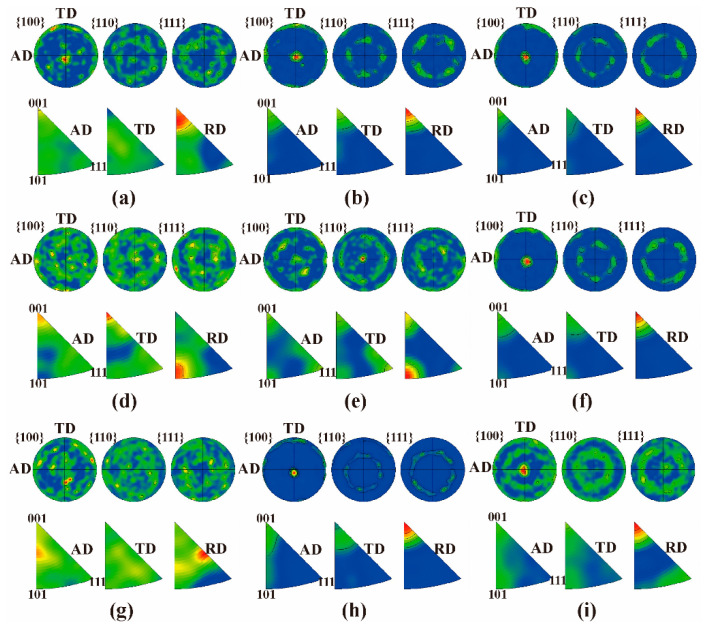
EBSD-derived positive and reverse polar diagrams of Cr coatings: (**a**–**i**) corresponding to samples # 1–9, respectively.

**Figure 8 materials-15-07177-f008:**
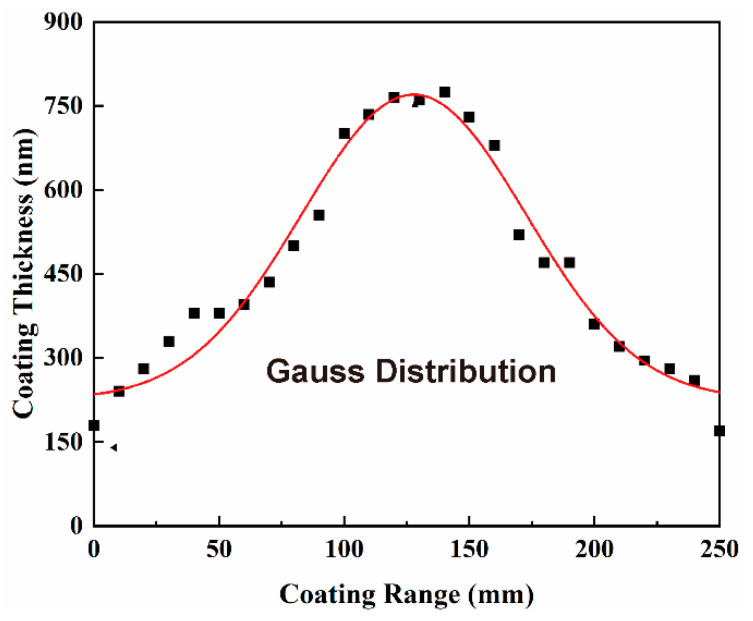
A Gauss distribution of coating thickness within a single target deposition range due to the “ion shrinkage” effect.

**Figure 9 materials-15-07177-f009:**
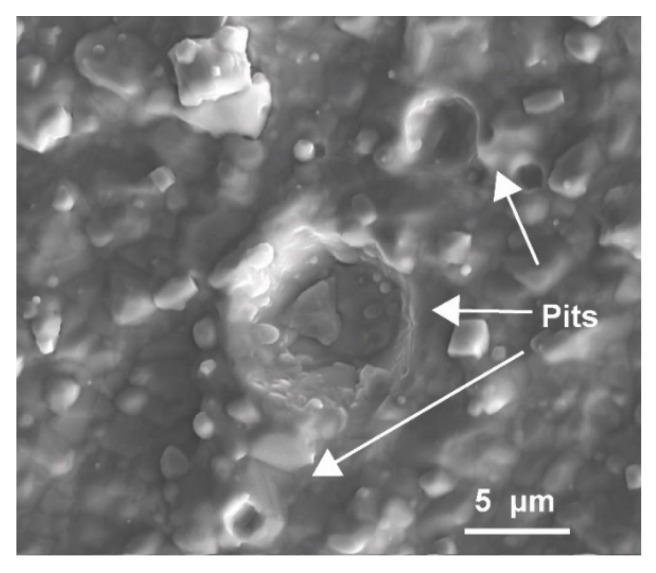
Morphology of pits left after removal of particles from the surface of Cr coating.

**Figure 10 materials-15-07177-f010:**
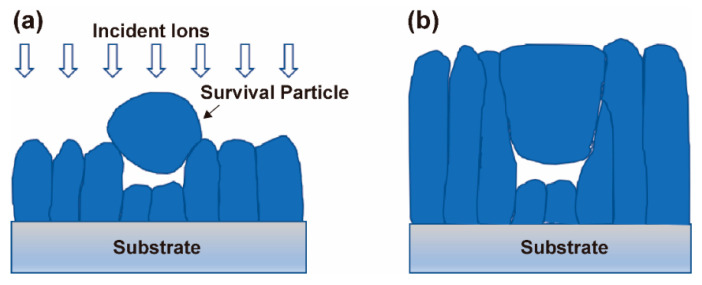
Schematic diagram of arcuate pore formation in Cr coating: (**a**) shadowing effect and (**b**) an arcuate pore left inside the coating.

**Table 1 materials-15-07177-t001:** L9(3^4^) orthogonal experiment parameters during steady-state deposition.

Test No.	Heating Temperature (°C)	Arc Current (A)	Gas Pressure (Pa)	Negative Bias Voltage (V)
1	350	150	1.6	−160
2	350	120	1.2	−120
3	350	90	0.8	−80
4	300	150	1.2	−80
5	300	120	0.8	−160
6	300	90	1.6	−120
7	250	150	0.8	−120
8	250	120	1.6	−80
9	250	90	1.2	−160

**Table 2 materials-15-07177-t002:** Statistical results of the coating thickness uniformity for a representative Cr-coated N36 tube (sample #8) (thickness unit: μm).

	Circum.	a	b	c	d	e	$\bar{x}$	*RAD*
Axial								
**1**		24.78	21.36	23.26	23.06	22.3	23	3.91%
**2**		27.05	24.86	26.86	26.29	25.34	26.1	3.01%
**3**		27.91	26.39	27.25	27.24	27.15	27.2	1.23%
**4**		27.06	24.34	25.73	24.88	24.78	25.4	3.27%
**5**		23.91	22.03	22.78	22.5	22.21	22.7	2.32%
** $\bar{x}$ **		26.1	23.8	25.2	24.8	24.4	24.9	2.54%
** *RAD* **		4.34%	6.84%	4.29%	5.81%	6.12%	5.45%	

**Table 3 materials-15-07177-t003:** Results of the range and variance calculations of average deposition rate.

Factor Level	Temperature (°C)	Arc Current (A)	Gas Pressure (Pa)	Bias Voltage (V)
*k* _1*j*_	2.196	** 3.297 **	2.408	2.188
*k* _2*j*_	** 2.513 **	2.222	2.239	2.321
*k* _3*j*_	2.357	1.547	2.419	** 2.557 **
*R_j_*	0.317	1.750	0.180	0.369
*SS_j_*	0.151069	4.673750	0.061105	0.209813

**Table 4 materials-15-07177-t004:** F-testing results of average deposition rate.

**Variance Source**	$SS_{j}\;$	$df_{j}\;$	$MS_{j}$	F	Table Value	Significance Level
Arc current	4.673750	2	2.336875	76.49	F_0.013_(2,2) = 75.92	α ~ 0.013
Bias voltage	0.209813	2	0.104906	3.43	F_0.226_(2,2) = 3.42	α ~ 0.226
Temperature	0.151069	2	0.075534	2.47	F_0.288_(2,2) = 2.47	α ~ 0.288
Error	0.061105	2	0.030552			
Sum	0.1791	8				

**Table 5 materials-15-07177-t005:** Results of the statistical analysis of surface droplet particles on the surface of Cr coatings obtained by Image J software.

Sample	Number	Diameter (μm)		Total Area (%)
		Ave.	Max.	
1	4717	1.00	8.96	20.00
2	3033	0.82	2.84	5.28
3	1079	0.95	3.74	2.57
4	4457	1.16	4.62	15.60
5	2107	0.96	4.59	3.84
6	739	0.93	2.96	1.29
7	4478	1.16	4.39	15.81
8	594	0.87	2.27	1.18
9	908	0.35	4.13	1.47

**Table 6 materials-15-07177-t006:** Calculated $TC_{hkl}$ values of orthogonal samples.

No.	1	2	3	4	5	6	7	8	9
TC_110_	1.803	1.812	1.461	1.911	1.134	2.136	1.599	1.797	1.407
TC_200_	0.567	0.654	0.747	0.537	0.864	0.588	0.615	0.570	** 1.026 **
TC_211_	0.630	0.537	0.792	0.552	** 1.005 **	0.276	0.786	0.633	0.567

**Table 7 materials-15-07177-t007:** Statistical results of average diameter ($\bar{d}$) and aspect ratio ($\bar{L/W}$) of Cr grains.

**No.**	Average		Maximum	
	$\bar{d}\left(\mu m\right)$	$\bar{L/W}$	$\bar{d}\left(\mu m\right)$	$\bar{L/W}$
1	1.548	3.302	7.471	14.225
2	1.795	4.353	6.189	18.211
3	1.527	4.370	4.882	17.193
4	1.709	4.178	8.801	20.793
5	1.713	4.031	6.926	18.438
6	1.708	4.297	4.814	19.061
7	1.589	3.519	8.705	16.091
8	** 2.044 **	4.111	** 11.890 **	** 22.843 **
9	** 1.094 **	3.197	** 3.215 **	** 12.426 **
mean	1.636	3.929	6.988	17.698

## Data Availability

The data presented in this study are available on request from the corresponding author. The data are not publicly available due to them being private property.

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
