# Peer review of "An Orthogonal Experimental Study on the Preparation of Cr Coatings on Long-Size Zr Alloy Tubes by Arc Ion Plating"

_materials, 2022, doi:10.3390/ma15207177_

Round 1

Reviewer 1 Report

Very good paper on the systematic study of the effect of arc ion plating parameters on the performance of Cr-coatings on long-length zirconium cladding tubes. The possibility of improving the characteristics of zirconium cladding tubes by applying Cr-coatings has long been known. However, this direction remains relevant due to the many different methods of applying such coatings, a large number of factors influencing the quality of coatings. Therefore, the work of the authors is original and relevant in the field of nuclear energy and it adds new scientific and production data for a specific coating method, namely arc ion plating. An impressive references list fully covers the problem. The use by the authors of a multivariate experiment with excellent statistical processing is the main point in favor. Discussion at a high scientific level. The conclusions consistent with the evidence and arguments presented and they address the main question posed. I have just a few comments on the paper:

1. Can the authors provide other dimensions of cladding tubes, namely diameter and wall thickness ?

2. Even though the English is not bad it can be improved due to a large number of typos and other errors. For example:

Line 18. ‘microstrutures’

Lines 82-83. ‘preffered’

Line 92. Lowercase letter in ‘large’

Line 120. ‘target-substate’

Line 134. ‘…voltages c and a constant…’ – unclear

Line 155. ‘depostion’

Line 189. ‘refered’

Line 231. Double ‘the’

Line 249. ‘depostion’

Line 284. ‘representive’

Line 341. ‘Futhermore’

Line 348. ‘caculations’

Line 409. ‘decreace’

Line 418. ‘stucture’

Line 445. Should be ‘calculations’

Line 474. ‘experimnetal’

Line 536. ‘explaination’

Line 551. ‘synchrouous’

Line 557. ‘ethier’

Line 647. ‘depositiing’

Line 666. ‘simutaneously’

Line 717. ‘shoud’

Line 747. ‘monotonicly’

Line 795. ‘temperture’

Line 837. ‘monotonical’

Line 842. ‘embeded’

Line 846. ‘prevailling’

Reviewer 2 Report

Dear Author,

I have read your manuscript with great attention and interests. Overall, the manuscript is well written. The introduction is relevant and results are clear and well explained. There are a number of comments and suggestion to improve the manuscript. After reviewing the manuscript, I recommend that this paper be accepted after minor revision.

Coating preparation

Line 46: Regarding process parameters under study, you clearly explain the selection of the temperature values for these experiments, but only mention that the rest of the values are considered from previous AIP data. Could you give us some kind of information to understand how did you proceed for the selection of arc currant, pressure, and Negative voltage? In addition, considering the conclusion, what is the reason for including gas pressure in this study?

Results

Line 318: The effect of heating temperature should not be ruled out in this conclusion.

Table 9: Could you explain the reason for obtaining a SSj value much higher than Kij values in the case of particle numbers?

Figure 10. Are the error bars shown in this Figure representing the experimental error of the measurement?

Conclusion

From your results and conclusions, it might be understood that arc current is the main parameter affecting coating process. Is there any explanation for the effect of the rest of parameters under study?

Reviewer 3 Report

Attached
